# Person-centred transition programme to empower adolescents with congenital heart disease in the transition to adulthood: a study protocol for a hybrid randomised controlled trial (STEPSTONES project)

Mariela Acuña Mora,[1,2] Carina Sparud-Lundin,[1] Ewa-Lena Bratt,[1,3] Philip Moons[1,2]

For numbered affiliations see end of article.

Correspondence to
Professor Philip Moons;
philip.moons@kuleuven.be

## ABSTRACT

**Introduction:** When a young person grows up, they evolve from an independent child to an empowered adult. If an individual has a chronic condition, this additional burden may hamper adequate development and independence. Transition programmes for young persons with chronic disorders aim to provide the necessary skills for self-management and participation in care. However, strong evidence on the effects of these interventions is lacking. Therefore, as part of the STEPSTONES project (Swedish Transition Effects Project Supporting Teenagers with chrONic mEdical conditionS), we propose a trial to assess the effectiveness of a structured, person-centred transition programme to empower adolescents with congenital heart disease in the transition to adulthood.

**Methods/design:** STEPSTONES will use a hybrid experimental design in which a randomised controlled trial is embedded in a longitudinal, observational study. It will be conducted in 4 paediatric cardiology centres in Sweden. 2 centres will be allocated to the randomised controlled trial group, assigning patients randomly to the intervention group (n=63) or the comparison group (n=63). The other 2 centres will form the intervention-naïve control group (n=63). The primary outcome is the level of patient empowerment, as measured by the Gothenburg Young Persons Empowerment Scale (GYPES).

**Ethics and dissemination:** The study has been approved by the Regional Ethical Board of Gothenburg, Sweden. Findings will be reported following the CONSORT statement and disseminated at international conferences and as published papers in peer-reviewed journals.

**Trial registration number:** NCT02675361; pre-results.

### Strengths and limitations of this study

- This is a multicentre study that uses a novel study design to test the effectiveness of a person-centred transition programme.
- Although several guidelines have recommended the implementation of transition interventions, the evidence of these is still insufficient. The study is expected to fill a gap in current transitional care knowledge.
- Owing to the type of intervention, blinding of participants is not possible.
- In order to have a long follow-up, transition interventions are recommended to start before the age of 16 years, which is not feasible in this study.

in diagnostic techniques and treatments, survival has increased, and 90% of them now reach adulthood.[2] Growing up with a chronic condition poses an extra burden on the individual, over and above the challenges typically related to adolescence. Management of the transition from childhood to adulthood and the shift from paediatric to adult care requires special attention from healthcare professionals. The movement to adult care is known as transfer, defined as "an event or series of events through which adolescents and young adults with chronic physical and medical conditions move their care from a pediatric to an adult health care environment".[3] This transfer should be preceded by a preparatory transitional phase. In healthcare, transition can be defined as the "process by which adolescents and young adults with chronic childhood illnesses are prepared to take charge of their lives and their health in adulthood".[3] To improve and facilitate this process, transition

## BACKGROUND

In developed countries, 10–40% of the adolescent population has a chronic disease with childhood onset.[1] As a result of improvements

programmes have been developed.[4] A well-designed transition programme is individually tailored, flexible, and age and developmentally appropriate.[5] The goal of transition programmes is to support the person's adjustment to a new care culture, to provide the necessary tools for self-care and to help satisfy medical, psychosocial and educational needs.[6] Evidence on the effects of transition programmes is scant, the level of existing evidence is low and the effectiveness of these interventions has yet to be determined.[7] This is due to the limited number of effectiveness studies; methodological issues (eg, contamination of control groups has not been accounted for in prior studies) and a short follow-up period of 12 months maximum.[7]

Patient empowerment is an asset that relates to the adolescents' self-management and participation in care.[8] Empowered adolescents can interact better with the adult healthcare system, and are more independent and more actively involved in decisions regarding their health.[9 10] However, patient empowerment has not been a formal target in transition programmes so far. To increase the level of empowerment, a person-centred care (PCC) approach is important.[11] This perspective allows for a partnership between the adolescent and healthcare providers in which the adolescent's views guide the intervention.[12] Through this approach, it is possible to share helpful resources and information with the adolescents and give them the opportunity to assume an active role in the decision-making process.[11]

Congenital heart disease (CHD) is a typical example of a life-course disease and can therefore be considered a chronic condition. CHD has a prevalence of 9.1 per 1000 newborns and represents 28% of all congenital defects.[13] Around 90% of children born with CHD reach adulthood,[2] and ~6.2 in 1000 adults live with this condition.[14] Although several centres have established a transition programme for individuals with CHD,[15] only one study on the effectiveness of such a programme has been undertaken so far. That study was conducted in Canada and used a quasi-experimental design to test the effects of a 1-hour intervention led by a nurse.[16] Significant improvements in the level of knowledge and self-management among patients were observed.[16] In addition, a comparative and a longitudinal study in Belgium confirmed the effectiveness of patient education.[17 18]

To address the gaps in evidence on the effectiveness of transition programmes, we established the STEPSTONES project (Swedish Transition Effects Project Supporting Teenagers with chrONic mEdical conditionS). This project aims at developing and evaluating the effectiveness of transition programmes for young persons with chronic childhood-onset conditions. Although the project will encompass different chronic conditions, it will first study adolescents with CHD.

## Objective and hypothesis

This article describes the protocol for a study to test the effectiveness of a transition programme to empower adolescents with CHD in the transition to adulthood and the transfer to adult care. The main hypothesis is that adolescents with CHD who receive a structured, person-centred transition programme over a 2-year period have a higher patient empowerment score than adolescents receiving usual care.

This study is presented in accordance with the guidelines issued by SPIRIT for reporting of study protocols.[19]

## METHODS/DESIGN
### Study design and setting
A hybrid experimental design will be used, in which a randomised controlled trial (RCT) is embedded in a longitudinal, observational study. This will result in a three-arm design (see figure 1). The study will be conducted in four CHD centres in Sweden. Two of the centres are allocated to the RCT, where participants are randomly assigned to either the intervention group or the comparison group. The other two centres are designated as the control group and represent intervention-naïve centres.

This type of design allows for the investigation of the effectiveness of the transition programme while also taking into account potential contamination of the comparison group in the intervention centres.

### Sample size estimation
Sample size calculation is based on the primary outcome of patient empowerment. The target is an improved patient empowerment score of 5.25 points on a scale from 15 to 75 (ie, 0.5 SD). For two-sided tests with $\alpha=0.05$ and power=80%, 63 patients are needed in each arm of the RCT. In order to compensate for a potential 10% dropout rate, we will recruit 70 patients in each arm of the RCT. An additional 70 patients will be recruited in the centres of the control group. Among the four centres, a total of 210 patients will be enrolled (see figure 1).

### Participants and recruitment
Eligible participants are (1) adolescents with CHD, defined as "a gross structural abnormality of the heart or the intrathoracic great vessels that is actually or potentially of functional significance";[20] (2) age 16 years; (3) Swedish speaking and (4) literate.

Patients are excluded if they do not have the physical or cognitive capacity to complete the questionnaires, or have acquired heart conditions or heart transplantation. Parents of the adolescents will also be asked to participate. All eligible participants are recruited by a transition coordinator (TC) or a data collection officer (DCO).

### Randomisation
Using a web-based system, a stratified block randomisation with a random variable block size will assign patients of the RCT centres to the intervention group or the comparison group (1:1).[21] Block randomisation

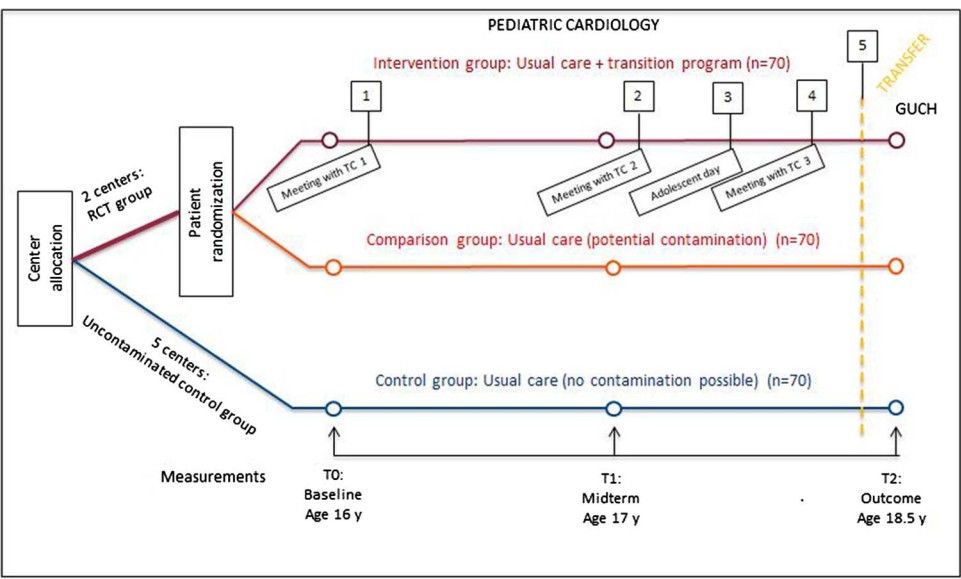

**Figure 1** Overview of the study design and implementation steps.

ensures that the TCs in the two centres have a relatively continuous exposure to the intervention over time, which allows them to keep their skills up to date and to ensure the fidelity of the intervention. The fact that the block sizes vary randomly minimises the possibility that the TCs could predict group assignments.

Stratification is made by centre and disease complexity. The stratification for centre ensures that the two centres have an equal number of intervention patients. Stratification by disease complexity (mild; moderate; complex)[22] is important to avoid over-representation of those people with, for example, complex CHD in one or the other group due to failing randomisation.

Overall, this approach decreases the within-centres variability and reduces the risk for bias and confounding.[21] The TCs are responsible for accessing the randomisation system in order to determine patient allocation.

## Intervention
### Transition programme (intervention group)
This study implements a multicomponent intervention, adapted from a brief transition programme developed by Hilderson et al.[23] Adaptation of the programme followed the intervention mapping process.[24] Intervention mapping is a thorough method used to build comprehensive programmes for health promotion and consists of six consecutive steps: (1) assessment of the needs of the target population; (2) defining programme objectives by specifying performance and change objectives; (3) determining theory-based intervention methods and practical strategies; (4) designing and producing the intervention programme; (5) adoption and implementation of the intervention; and (6) developing and executing a plan for evaluation.[24] Applying the intervention mapping method in this study ensures that the

intervention is well-grounded and includes the necessary components to target the programme objectives. A detailed description of each of the steps will be given in a related article.

The transition programme includes eight key components: (1) a TC; (2) a written person-centred transition plan; (3) provision of information and education about the condition, treatment and health behaviours; (4) availability by telephone and email; (5) information about and contact with the Grown-Up Congenital Heart Disease (GUCH) clinic; (6) guidance of parents; (7) meeting with peers; and (8) the actual transfer to the GUCH clinic.

These components are implemented in five steps: (1) a first patient visit with the TC; (2) a second outpatient visit with the TC; (3) information day for adolescents and their parents; (4) a third outpatient visit with the TC; and (5) actual transfer to the GUCH clinic. The TCs are specialised nurses at the outpatient clinic of paediatric cardiology who have received specific training in performing this intervention.

Throughout the intervention, the TCs employ a PCC approach. PCC implies that participants are active agents in their care and decision-making process.[12] Ekman et al[12] proposes that PCC can be implemented in three stages. First, initiating the partnership by allowing the patients to tell their story (patient narratives). This allows for the patients' views to be at the centre of care. Second, developing the partnership through constant communication between the patient and healthcare providers. This is accomplished by sharing experiences and building a common understanding of the care plan. Third, safeguarding the partnership by documenting the decisions made and the preferences of the patients.

The content of the visits is individualised, age and developmentally adapted, and it is documented in the

individualised transfer plan. The TCs, together with the adolescent, are in charge of determining which topics are important to discuss, in establishing goals related to patient empowerment and in assessing the need for referral to other services.

### Usual care (intervention, control and comparison group)

All adolescents who participate in the study receive usual care, irrespective of whether in the intervention, comparison or control group. Usual care includes visits to an outpatient paediatric cardiology clinic where they meet a nurse and a paediatric cardiologist. Disease complexity determines the frequency of the visits. None of the participating centres have a TC as a part of usual care and there is no documentation of transition plans. It is common practice in Sweden that patients are transferred to adult care around the age of 18 years.

Since usual care can vary across clinics and over time, an assessment of transition-related practices will be carried out in all the centres at the beginning of the intervention and at the end of the study. This assessment will comprise three elements: (1) a structured survey called CHAT (Congenital Heart Adolescent and Transition; on file); (2) interviews with healthcare professionals who are involved in the care for adolescents with CHD in the participating centres and (3) observations of the routine during the clinic visits.

### Blinding

Owing to the nature of the intervention and the study design, it is not possible to blind the participants, TCs or DCOs. However, the TCs and DCOs in charge of carrying out the intervention and data collection are not involved in the preparation of the intervention design or in the statistical analyses.

To avoid cross-contamination between the comparison and the intervention group within the RCT centres, TCs are not involved in the delivery of care of the comparison group and other healthcare professionals in the participating centres will not be informed of the intervention components. DCOs in the control group do not participate in discussions of the intervention.

### Intervention fidelity

To ensure that the transition programme is implemented according to the developed intervention, the TCs are selected based on their experience of working with children and adolescents with CHD. In preparation for this study, they received a specifically designed 4 days training programme, addressing adolescents' health and development; communicating with and interviewing young persons; patient education; PCC; and the detection of and screening for risk behaviours. Training is given by experts in the aforementioned fields and features different pedagogical approaches, for example, group discussions, case assessments and role play. In addition, the TCs were asked to read two textbooks on adolescent health and PCC.

Standard operating procedures (SOPs) for each group of the study were developed in order to ensure consistency throughout the study. These SOPs included: ethical approval; study preparation; study administration; participants recruitment and survey completion; intervention characteristics; usual care; data storage, entry and security; study progress; and publication policy.

Intervention fidelity is monitored as part of the process evaluation. The Medical Research Council (MRC) guidance on process evaluation of complex interventions is applied,[25] and both quantitative and qualitative methods will be used. In the quantitative assessment, we investigate the acceptability, adherence/fidelity and attrition, as well as potential safety issues. These data are collected using three assessment forms (enrolment/follow-up form, adverse events (AEs) report and intervention implementation form) filled out by the TCs during the intervention period. The qualitative assessment will obtain information about the experiences of and implementation by the TCs and will provide insights into the mechanisms of impact based on the experiences of patients and their parents. The process evaluation will be undertaken as a separate study conducted in parallel with this effectiveness study.

### Follow-up period and data collection

During the 2-year follow-up period, outcome measurements are assessed on three different occasions: T0, baseline when the participants are 16 years old; T1, midterm when the participants are 17 years; and T2, final data collection when the participants have been transferred to adult care and are ~18.5–19 years old (see figure 1).

Participants from the intervention group will be mailed the set of questionnaires 1 month before the outpatient clinic visit. If the questionnaires have not been received by the time of the visit, they will have the opportunity to complete them while waiting for their appointment. Patients from the control and comparison groups may not have a scheduled outpatient visit. Therefore, data collection in these groups is completely undertaken by post when the participants are 16 (T0), 17 (T1) and 18.5 years old (T2).

To minimise non-response in the comparison and control groups, a modified Dillman procedure will be used.[26] Two weeks after the documents are sent, non-responders will receive a first personalised reminder in the form of a text message. If after 4 weeks the questionnaires have not been received, a reminder letter and a new set of questionnaires is sent. After 6 weeks, persistent non-responders are contacted by telephone and asked whether they received the questionnaires, if the mailing address is correct, if they want a new set of questionnaires and if they want to continue participating in the study. After 8 weeks, a last reminder is sent to those who agree to participate in the study during the telephone call.

## Outcome measurements

A background information questionnaire is included, comprising sociodemographic variables such as sex, age, educational level, size of community, place of birth, number of siblings, birth order, living situation, other health conditions and medication (obtained from self-report and medical file).

Standardised questionnaires are used to assess primary and secondary outcomes. Table 1 presents an overview of the characteristics and psychometric properties of the questionnaires used.

### Primary outcome measures

The primary outcome is patient empowerment, defined as "an enabling process or outcome arising from communication with the healthcare professional and a mutual sharing of resources over information relating to illness, which enhances the patient's feelings of control, self-efficacy, coping abilities and ability to achieve change over their condition".[9] Empowerment includes five dimensions: (1) identity; (2) knowledge and understanding; (3) personal control; (4) decision making; and (5) enabling others.[9]

Since no questionnaire to measure the level of empowerment in young persons with chronic conditions exists, we developed the Gothenburg Young Persons Empowerment Scale-Congenital Heart Disease module (GYPES-CHD). The scale has 15 items, 3 items for each of the aforementioned dimensions. Each item is answered on a five-point Likert scale: strongly disagree; disagree; neither agree nor disagree; agree and strongly agree. A total score is calculated by summing the item scores. The total score ranges from 15 to 75 points, with a higher score reflecting a higher level of patient empowerment.

Face validity was confirmed after cognitive interviews with six young persons with CHD. The psychometric properties of this scale were examined in a cross-sectional study in 205 young persons with CHD. Internal consistency of the items was confirmed with a Cronbach's α of 0.821. No floor or ceiling effect in scoring was observed. A complete description of the psychometric evaluation of GYPES will be provided in another article.

### Secondary outcomes measures

We will measure six secondary outcomes in the adolescents: transition readiness, health behaviours, knowledge about CHD, healthcare usage, quality of life and patient-reported health. In the parents, we will measure transition readiness (proxy) and parental uncertainty towards transition.

### Transition readiness

Transition readiness is defined as the "adolescent's readiness to assume complete responsibility for their healthcare and their readiness to transfer to adult medical care".[27] Transition readiness is evaluated using the Readiness for Transition Questionnaire (RTQ), adolescent and parent versions. These questionnaires examine two aspects. First, the overall transition readiness is assessed, using two items ranging from 1 to 4. The sum of these two items results in a score ranging from 2 to 8. Second, the frequency of adolescent responsibility and parental involvement are reported in 10 different health behaviours, on a five-point Likert scale. Each variable results in a total score ranging from 10 to 40, and higher scores indicate higher adolescent responsibility or parental involvement, respectively.

### Knowledge about CHD

Adolescents' knowledge about their CHD is assessed with the Knowledge Scale for adults with Congenital Malformed Hearts (KnoCoMH).[28] This instrument evaluates the level of knowledge within four domains: disease and its treatment; complications prevention; physical activity; and reproductive issues.[28] The questionnaire includes 19 items and scores are calculated by dichotomising answers as correct or incorrect. Access to the medical files is needed in order to confirm the answers in some of the items. For participants for whom medical treatment or endocarditis prophylaxis is not recommended, the domains are excluded.

### Health behaviours

Health behaviours are measured with the Health Behavior Scale-CHD. Health behaviours are 'activities a person undertakes to maintain or improve health and prevent diseases'.[30] This scale assesses alcohol consumption, tobacco use, dental care and physical activity. The scale has 15 items that help to calculate three summary risk scores: substance use (score 0–100), dental hygiene (score 0–100) and total health risk score (score 0–100). Higher risk scores represent unhealthier behaviours.

### Patient-reported health

Patient-reported health is assessed by the generic and cardiac modules of the Pediatric Quality of Life inventory 4.0 (PedsQL 4.0), as well as the EuroQoL 5 dimensions 3 levels (EQ-5D-3L). The PedsQL 4.0 provides information on health status in relation to psychosocial and physical health.[32] The generic module consists of 23 items that cover four dimensions: physical; emotional; social functioning and school functioning. Items are answered by using a five-point Likert scale: never, almost never, sometimes, often and always. For each dimension, a scale score can be calculated, which ranges from 0 to 100. Further, a physical health summary score, a psychosocial health summary score and a total score can be calculated.

The cardiac module contains 27 items that cover six dimensions: symptoms; treatment; perceived physical appearance; treatment anxiety; cognitive problems and communication. Each item is answered using a five-point Likert scale (never, almost never, sometimes, often and always). As in the generic version, scores are

**Table 1**  Psychometric properties of the questionnaires used in STEPSTONES-ConHD

| Variable | Subject | Measurements | Items | Validity | Reliability | Responsiveness | Interpretation |
|---|---|---|---|---|---|---|---|
| **Primary outcome** | | | | | | | |
| Patient empowerment | AD | GYPES | 15 | Construct validity supported an adequate model fit of a five factor solution (df: 80; $\chi^2$: 154–948, p<0.0001; CFI: 0.916; RMSEA: 0.068; SRMR: 0.069) (data on file) Face validity confirmed in young persons with chronic conditions (data on file) | Internal consistency confirmed in young persons with CHD, $\alpha$=0.819 (data on file) | Floor scores: 0%; ceiling scores: 1.5% (data on file) | Score from 15 to 75. Higher score reflects a higher level of patient empowerment |
| **Secondary outcomes** | | | | | | | |
| Transition readiness | AD | RTQ adolescent version | 26 | Validity based on relationships with other variables confirmed in young persons with kidney transplant,[27] p.91–92 | Internal consistency confirmed in young persons with kidney transplant; $\alpha$ values over 0.70[27] | NR | Scores from 10 to 40. Higher scores denote increased adolescent or caregiver responsibility. |
|  | PA | RTQ parent version |  | Construct validity reported for parents of young persons with kidney transplant,[27] p.91–92 | Reliability reported for parents of young persons with kidney transplant; $\alpha$ values over 0.70,[27] p.90 | NR | |
| Knowledge on ConHD | AD | KnoCoMH | 19 | Validity was confirmed in adults with CHD in relation to discrimination ability, the relationship of the items and the construct of interest,[28] p.232–234 KnoCoMH is based on the Leuven Knowledge Scale which has been validated in young persons with CHD,[29] p.79–80 | Internal consistency and test–retest reliability confirmed,[28] p.232–234 | NR | Scores are calculated by dichotomising the answers (correct/incorrect) for each domain |
| Health behaviours | AD | HBS-CHD | 15* | Item content validity, scale content validity index and validity based on relationships to other variables confirmed in adolescents with CHD,[30] p.548 | Stability not confirmed[29] | Confirmed in adolescents with CHD by Guyatt's Responsiveness Index,[30] p.551 | Substance use, dental hygiene and total health risk score from 0 to 100. Physical exercise score from 0 to $\infty$ |
| Patient-reported health | AD | PedsQL generic module | 23 | Convergent validity, measurement invariance and factor structure confirmed in paediatric populations,[31] p.336–337 [32] p.1507–1510 | Test–retest reliability and internal consistency show $\alpha$ values over 0.70 in paediatric populations,[32] p.1507 [31] p.333–335 | Minimal clinically important difference reported for young people with CHD was 6.0 for the total score.[33] | Scores from 0 to 100. Higher scores indicate a better perceived health status. |

**Table 1** Continued

| Variable | Subject | Measurements | Items | Validity | Reliability | Responsiveness | Interpretation |
|---|---|---|---|---|---|---|---|
| | | PedsQL cardiac module | 27 | Convergent validity was confirmed,[34] p.215 | Internal consistency for majority of scales exceeded α values of 0.70,[34] p.214 | Minimal clinically important differences reported for young people with CHD ranged from 7.6 for symptoms to 12.6 for communication.[33] | Scores from 0 to 100. Higher scores indicate lower problems. |
| | | EQ-5D-3L | 5 | Convergent and discriminative validity confirmed in patients with cardiovascular disease,[35] p.6 | Test–retest reliability confirmed in cardiovascular patients,[35] p.6 | Confirmed in patients with cardiovascular disease,[35] p.6 | Scores from 1 (no problems) to 3 (extreme problems) on five dimensions Scores from 0 (worst imaginable health state) to 100 (best imaginable health state) |
| Quality of life | AD | LAS | 1 | Validity based on relationship with Satisfaction with Life Scale found to be highly correlated (ρ=0.52). Test content confirmed in adults with CHD (100% of patients understood the wording and format)[36] Validity has been evaluated in adolescents with CHD.[37 38] | Test–retest reliability confirmed in adults with CHD (ICC=0.65),[36] | Confirmed in adults with CHD (floor scores=0%, ceiling scores=2.7%)[36] | Score is from 100 (best imaginable health status) to 0 (worst imaginable health status). |
| Healthcare usage | AD | Healthcare usage questionnaire | 11 | NA | NA | NA | NA |
| Parental uncertainty towards transfer to adult care | PA | LAS | 1 | Face validity confirmed in parents of young persons with chronic conditions (data on file). Test content confirmed in parents of young persons with CHD (100% of patients understood the wording and format). | NR | Floor scores: 7.8%; ceiling scores: 4.8% (data on file) | Score is from 0 (very uncertain about the transfer to adult care) to 100 (not uncertain about the transfer of care). |

AD, adolescents; CFI, comparative fit index; CHD, congenital heart disease; EQ-5D-3L, EuroQoL 5 dimensions 3 levels; GYPES, Gothenburg Young Persons Empowerment Scale; HBS-CHD, Health Behavior Scale-Congenital Heart Disease; ICC, intra-class correlation; KnoCoMH, Knowledge Scale for Adults with Congenitally Malformed Hearts; LAS, linear analogue scale; NA, not applicable; NR, not reported; PA, parents; PedsQL, Pediatric Quality of Life Inventory 4.0; RMSEA, root mean square error of approximation; RTQ, Readiness for Transition Questionnaire; SRMR, standardised root mean square residual; STEPSTONES, Swedish Transition Effects Project Supporting Teenagers with chrONic mEdical conditionS.

transformed on a scale from 0 to 100, and then a mean subscale core is calculated.[34]

The EQ-5D-3L comprises a descriptive system and a visual analogue scale (VAS). The descriptive system has five dimensions (mobility, self-care, usual activities, pain/discomfort and anxiety/depression) with three response categories per dimension.[39] The participants are asked to indicate which statement is most suitable in relation to their health status. The VAS measures the participants' self-rated health on a scale with the end points 'best imaginable health possible' (=100) and 'worst imaginable health state' (=0). The descriptive system allows for the determination of a five-digit health state that combines the level of perceived problems in each dimension (level 1: indicating no problem, level 2: indicating some problems and level 3: indicating extreme problems). Information obtained from the descriptive system can be converted to a single summary index by assigning values to each level of every dimension.

### Quality of life

Quality of life is defined as "the overall degree of life satisfaction that is positively or negatively influenced by individuals' perception of certain aspects of life important to them, including matters both related and unrelated to health".[40] This variable is measured using a 10 cm linear analogue scale (LAS),[36] where the participants indicate in the scale how they judge their current quality of life. The scale looks similar to the VAS of the EQ-5D, but has the end points 'best imaginable quality of life' (=100) and 'worst imaginable quality of life' (=0).

### Healthcare usage

Healthcare usage within the preceding 6 months is measured in relation to their CHD and other possible conditions. The measurement consists of six dimensions: hospital stay; healthcare contact; specialised care; emergency care; school absence and follow-up unit after transfer to adult care. This assessment was developed to be used in the APPROACH-IS project.[41] After the transfer to adult care, we will assess the proportion of patients who are in follow-up in a GUCH unit, both in the intervention and control arms.

### Parental uncertainty

Parental uncertainty is understood as the degree of certainty/uncertainty the parents have in relation to their children being transferred to adult care. This outcome is measured through LAS with the end points 'extremely uncertain' (=100) and 'not uncertain at all' (=0). This LAS has been developed for the purpose of the present study.

In addition to the primary and secondary end points, we will calculate a composite end point. We will do so because the transition programme is a complex intervention that may have varying effects on different outcomes. Prior studies on the impact of transition programmes showed that the intervention has impacted on transition readiness,[16] patient knowledge,[16] quality of life[42] and psychosocial health.[42] Therefore, each patient will be categorised as improved or not. Patients will be classified as improved if they have an improvement in empowerment, transition readiness, knowledge, quality of life or psychosocial health of 0.5 SDs compared with their baseline measurement.

### Data management

Data are entered into a web-based system by the TC and the DCO. Those with access to the system are the TC, DCO, project coordinator and principal investigator, each of whom has a personal login. All data are checked for completeness and the quality of the data is monitored at least twice a year by the members of the research group. This process is carried out by verifying records with the raw data and checking frequency distributions to identify responses outside of the given possibilities.

### Statistical analyses plan

The primary analyses will be performed following the intention-to-treat principle which helps to avoid biases in the comparisons between groups. The primary end point (level of empowerment) will be analysed using Fisher's non-parametric permutation test unadjusted between the two groups. Complementary between-group analyses will be made using analysis of covariance adjusted for baseline variables that differ significantly between groups and for each centre.

All analyses will be predefined in a comprehensive statistical analysis plan before data lock. The level of significance will be at $p \leq 0.05$. Two-sided tests will be used.

### Trial duration

Patient recruitment began in July 2016 and is planned to finish in December 2017. Once recruited, patients are followed up until they are transferred to GUCH. Data collection is expected to be finished by December 2019. After the data set is locked, analysis and dissemination of results will begin.

### Ethical considerations

Before inclusion in the study, the TCs and DCOs will seek written informed consent from the adolescents and their parents after they have received written and oral information about the study. Participants have the right to withdraw at any point throughout the study. The principles established in the Declaration of Helsinki will be followed during the entire duration of the study.[43]

To ensure confidentiality, personal data are coded. Participant and study-related information is stored in locked cabinets and digital documents are password-protected.

### Follow-up on AEs

All AEs must be reported to the principal investigator and project coordinator within a week. In case of a

serious AE, the report must be made immediately. Appropriate measures will be undertaken in every AE in order to ensure patient safety.

## Dissemination

The results of this study will be submitted to peer-reviewed journals for publication following the CONSORT statement.[44] Abstracts for poster or oral presentations will be submitted to international conferences. Authorship is granted to a collaborator from every participating centre and members of the research group fulfilling the criteria of authorship.[45 46] If possible, we will include 'on behalf of STEPSTONES-CHD' in the list of authors. After completion of the study, the data set will be available on formal request.

## DISCUSSION

Previous studies have tried to determine the effectiveness of transition programmes. However, owing to several methodological limitations, the true effect of these interventions is still uncertain.[7] Therefore, we will conduct a study that assesses the effectiveness of a structured, person-centred transition programme to empower adolescents with CHD in the transition to adulthood. Through the implementation of the proposed transition programme, we expect an increase in the level of empowerment. A high level of empowerment can help the adolescent navigate through the adult care system, participate in care planning, and increase self-management and decision-making skills.[8]

The intervention in this trial was designed and initially evaluated by Hilderson et al[23 42] in Belgium. Therefore, it was important to determine if the intervention components were suitable in the Swedish setting. To accomplish this, five preparatory studies were carried out (four qualitative, one quantitative study). The qualitative studies included individual and focus group interviews with young persons with CHD and their parents.[47 48] These studies provided an overview of the current transition practices in Sweden and the needs of this population before the transfer to adult care. The quantitative study was a cross-sectional study including 205 young persons with CHD, and served as a pilot-test for our primary outcome measurement. Furthermore, the intervention was discussed with a panel of international experts with experience in transition research in adolescents with chronic conditions and patient representatives.

The preparatory studies and the discussion with the panel of international experts improved the quality, feasibility and acceptability of the intervention. This interdisciplinary and comprehensive approach increases the chance of producing a high-quality intervention with positive results.[49]

This study has several strengths, because we explicitly take limitations of prior studies into account. First, we employ an RCT which is the gold standard for the assessment of the effectiveness of interventions and provides the highest level of evidence. The stratified block randomisation that we undertake ensures that the intervention and comparison groups are balanced and that potential confounding factors, such as complexity of the disease or centre characteristics, are equally distributed and therefore minimise the risk of biasing the results.

Second, the proposed hybrid experimental design allows us to account for potential contamination of the control group. Indeed, the three arms allow performance of direct comparison between the intervention and comparison group while enabling a comparison of findings with those obtained in intervention-naive centres.

Third, patient empowerment is used as the primary outcome. Empowerment is argued to be important for the development and application of skills to promote behavioural changes in the adolescents.[10] Admittedly, empowerment can be seen both as a process and an outcome.[10] However, being an intermediate outcome, empowerment can be argued to be proximal to the intervention, whereas continuing cardiac care, reduced morbidity or mortality are probably the ultimate, but more distal outcomes.

Fourth, the PCC approach that is used throughout the transition programme is an important asset in the process of empowering patients.[11] Providing transitional care from a person-centred perspective is novel and has not been evaluated before in this age group.

Fifth, we follow-up the patients for more than 2 years. While doing so, we substantially expand the follow-up period of 12 months as used in prior studies.[7]

Sixth, since usual care may vary across centres and over time, we carry out a formal usual care assessment. Such a usual care assessment is important to draw firm conclusions, and yet this is seldom performed in clinical trials.[50]

Irrespective of these strengths, we anticipate some limitations as well. First, we are not able to blind the participants or the healthcare team that will perform the intervention. To minimise a potential impact, neither the therapeutic team nor the TCs participate in the data analysis. Second, there is a possibility of a relatively long period of recruitment for participants which can require extending the trial period and delaying data analysis. Third, transition interventions have been advised to start early on, for example, between 10 and 12 years of age, and have a long follow-up.[51] Furthermore, a long follow-up allows continuity of care to be measured in this population and provides insight into the effects of the transition programme over this parameter. Although we recognise the importance of starting early, it would have jeopardised the feasibility of the study, because it would have taken almost a decade to finish recruitment and follow-up until the age of 18 years. Nonetheless, it could be possible to later on assess loss of follow-up in a separate study.

## CONCLUSION

This article described the study protocol for a study to assess the effectiveness of a structured, person-centred transition programme for young persons with CHD. This study accounts for limitations of previous research and addresses an existing gap in the evidence of transitional care. It is hoped that the findings will provide strong evidence of the effectiveness of this intervention in young people with CHD. If successful, this transition programme could potentially be implemented as part of routine care provided to young persons with CHD, and transfer to other chronic conditions can be explored.

## Trial status

Status of the trial at the time of submission: patient recruitment.

**Author affiliations**
[1]Institute of Health and Care Sciences, University of Gothenburg, Gothenburg, Sweden
[2]Department of Public Health and Primary Care, KU Leuven, Leuven, Belgium
[3]Department of Pediatric Cardiology, The Queen Silvia Children's Hospital, Gothenburg, Sweden

**Acknowledgements** The authors would like to acknowledge the collaboration from the members of the panel of international experts and of the Advisory Board.

**Funding** This study is supported by research grants from the Swedish Heart-Lung Foundation (grant 20150535); Swedish Research Council for Health, Working Life and Welfare-FORTE (grant STYA-2015/0003); Swedish Children Heart Association; Swedish Research Council (grant 2015-02503); and the Institute of Health and Care Sciences of the University of Gothenburg.

**Competing interests** None declared.

**Ethics approval** The study was approved by the Regional Ethical Board of Gothenburg, Sweden (No.931-15).

**Provenance and peer review** Not commissioned; externally peer reviewed.

**Data sharing statement** Preliminary data on the psychometric properties of the GYPES and the LAS for parental uncertainty are on file. They will be integrated in a separate article on the psychometric evaluation of the instruments, which will be available in some months.

**Open Access** This is an Open Access article distributed in accordance with the terms of the Creative Commons Attribution (CC BY 4.0) license, which permits others to distribute, remix, adapt and build upon this work, for commercial use, provided the original work is properly cited. See: http://creativecommons.org/licenses/by/4.0/

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
