## [Reviewer comments · BMJ Open]

ARTICLE DETAILS

TITLE (PROVISIONAL)	A person-centered transition program to empower adolescents with congenital heart disease in the transition to adulthood: study protocol for a hybrid randomized controlled trial (STEPSTONES project)
AUTHORS	Acuna Mora, Mariela; Sparud-Lundin, Carina; Bratt, Ewa-Lena; Moons, Philip

VERSION 1 - REVIEW

REVIEWER	MOCERI Pamela Cardiology department and Paediatric Cardiology department Pasteur University Hospital CHU de Nice FRANCE
REVIEW RETURNED	24-Oct-2016

GENERAL COMMENTS	This is very interesting study protocol. To date, this is the first attempt of randomized study investigating the role of formal transition protocol in CHD adolescents. Regarding the methodology, the hybrid concept of longitudinal cohort study including a randomized study is probably the more accurate way to study the effect of transition. The primary outcome is patient empowerment. It's therefore challenging to assess. In order to answer this question the authors have developed a score already assessed during a preliminary study. The authors should give the reference for this study. Secondary outcomes are the following: transition readiness, health behaviors, knowledge about ConHD, healthcare utilization, quality of life and patient reported health. The authors, as they aim to provide a full study about the impact of a transition programme should also assess patients for loss of follow-up. Actually one of the major problems is loss of follow-up of CHD Young adults. The duration of the study, 2 years, is probably not long enough to assess this item. However, the authors should consider to include this parameter in a future analysis, and mention it in the 1st protocol. Questionnaires to assess the outcomes have been carefully chosen and the literature has already validated those tools. The authors should consider to extend the FU of their patients, even if some issues can be anticipated.
---

REVIEWER	Emily Heery Oireachtas Library and Research Service Ireland
REVIEW RETURNED	02-Nov-2016

GENERAL COMMENTS**Abstract**

Lines 11-12 (p. 2): the word 'an' should be removed

Article summary

Line 25 (p. 3): should it be "programme" not "program"?

Lines 16-41 (p. 3): I'm not sure 'strengths and limitations' fully make sense as a subheading here. Some points are challenges rather than limitations (e.g., point 4)

Background

P.4 First paragraph – I think transfer should also be defined in the manuscript especially as you are making comparisons before and after transfer, so it is important to distinguish the concept of 'transition' from the concept of 'transfer'.

Lines 55-56 (p.4): "To achieve this..." this sentence appears to be unfinished.

Lines 10-11 (p. 5): CHD may be a better acronym

Lines 42/43 (p.5): "has a generic nature" maybe rephrase.

Methods/Design

Lines 47-57 (p.6) Does your sample size calculation consider potential drop-out. Would your sample size for the intervention group be large enough to take account of this?

Lines 32/33 (p. 12) Using coloured ink in questionnaires has also been shown to increase responses rates. So if they look appealing this may encourage a response

Lines 32/33 the word "rate" could be removed

Lines 32-48 (p. 12) I wonder if the time to first follow-up of non-responders is a little long at three weeks? I think a one or two week gap might be better in terms of achieving a high response rate. Also, as the project involves young people and mobile use is high in this group, a personalised text message or social media message may be an effective first reminder, that is not too intrusive.

Table 1. You use a mixture of child and adult questionnaires to measure outcomes. Are the questionnaires suitable for comparing young people and adults and are they all age appropriate at the time-points assessed? Do the participants fill out all questionnaires at each timepoint?

Timing of transfer: Is this standardised across clinics? Maybe the methods sections should include information on the usual timing of transfer at each clinic and how variation could potentially affect the intervention. For example, if some patients didn't transfer until 20 years of age or they transferred at 16/17 years of age.

P.18 -20 maybe some subheadings could be useful to break up the text in this section.

P.20 line 27 should be "transferred"

P. 21 lines 5/6 "TC" and "DCO" are these acronyms necessary?

P.22 line 48 Should be: "will be submitted to peer reviewed journals for publication"

Discussion

P. 23 line 22 Is there an 'and' missing after the word 'planning'?

P.23 line 37 – this should be 'group' rather than 'groups'

P.23 line 55 'improved' may be a better word than 'increased' here.

P.24 line 17 This should be 'potential' rather than 'potentially' I think

P.24 line 20 the end of this line should be rewritten perhaps 'risk of biasing' instead of 'risk for bias'

P.25 line 44 I think the word 'proves' here is far too strong. Also the word 'expected' is too strong. Possibly the sentence could be rephrased as:

"It is hoped the findings will provide strong evidence of the effectiveness of this intervention in young people with ConHD"

P.25 line 48 'can' should be changed to 'could potentially'.

VERSION 1 – AUTHOR RESPONSE

Reviewer 1

-To date, this is the first attempt of randomized study investigating the role of formal transition protocol in CHD adolescents.

Regarding the methodology, the hybrid concept of longitudinal cohort study including a randomized study is probably the more accurate way to study the effect of transition.

Response: We appreciate this positive comment from the reviewer. Thank you.

-The primary outcome is patient empowerment. It's therefore challenging to assess. In order to answer this question the authors have developed a score already assessed during a preliminary study. The authors should give the reference for this study.

Response: The Gothenburg Young Persons Empowerment Scale (GYPES) was developed following the conceptual framework proposed by Small (2013). The initial version of the scale was answered by 205 adolescents with congenital heart disease as part of a cross sectional study. This allowed us to determine the reliability based on internal consistency and the construct validity through confirmatory factor analysis.

These analyses showed good internal consistency and good model fit, but with low factor loadings in one of the items. In order to optimize the scale three items were rephrased.

Subsequently, this revised scale was evaluated in a group of adolescents with diabetes. The data obtained in this sample showed the scale is internally reliable and an adequate model fit was reached. This revised scale is the one being used in the transition program.

Currently, we are writing a dedicated article on the psychometric properties of GYPES. Therefore, at the moment reference for this study cannot be provided. Nonetheless, we have provided additional data in Table 1 on the results from the confirmatory factor analyses performed (p.13).

-Secondary outcomes are the following: transition readiness, health behaviors, knowledge about ConHD, healthcare utilization, quality of life and patient reported health. The authors, as they aim to provide a full study about the impact of a transition programme should also assess patients for loss of follow-up.

Actually one of the major problems is loss of follow-up of CHD Young adults. The duration of the study, 2 years, is probably not long enough to assess this item. However, the authors should consider to include this parameter in a future analysis, and mention it in the 1st protocol.

Questionnaires to assess the outcomes have been carefully chosen and the literature has already validated those tools.

The authors should consider to extend the FU of their patients, even if some issues can be anticipated. Indeed, one of the major problems this population faces is loss to follow-up and we appreciate the reviewer's comment.

Response: We could expect the transition program to improve the follow-up of these patients, in the sense they become aware of the need to continue under medical care in the Grown-Up Congenital Heart Disease unit (GUCH). We could hypothesize that 100% of the patients in the intervention group will be under GUCH follow-up at the age of 18.5y. It is likely to be lower in the comparison and control groups. This parameter is part of the healthcare utilization assessment that we conduct in the study.

The impact of the intervention on the lost to follow-up on the longer terms is indeed a pertinent issue. It is definitely our ambition to conduct a long-term follow-up study. Hence, we have added the following information in the discussion section.

The section now reads as:

“Third, transition interventions have been advised to start early on, e.g. between 10-12 years of age, and have a long follow-up. Furthermore, a long follow-up allows measuring continuity of care in this population and provide insight on the effects of the transition program over this parameter. Although we recognize the importance of starting early, it would have jeopardized the feasibility of the study, because it would have taken almost a decade to finish recruitment and follow-up until the age of 18 years. Nonetheless, it could be possible to later on, assess loss of follow-up in a separate study”

(p.26-27).

Reviewer 2

-Abstract

Lines 11-12 (p. 2): the word 'an' should be removed

Response: Typographical error has been corrected (p.2)

-Article summary

Line 25 (p. 3): should it be "programme" not "program"?

Response: Our manuscript was proof read for American English. "Programme" is usually used in British English. Thus, "program" is more appropriate to use. (p.3)

-Background

Lines 16-41 (p. 3): I'm not sure 'strengths and limitations' fully make sense as a subheading here. Some points are challenges rather than limitations (e.g., point 4) "Strengths and limitations" is the require subheading by the Journal in this section.

Response: We have removed point 4 "a long recruitment period may require extending the trial and delaying data analysis". This is indeed a challenge and not a limitation of the study (p.3).

-P.4 First paragraph – I think transfer should also be defined in the manuscript especially as you are making comparisons before and after transfer, so it is important to distinguish the concept of 'transition' from the concept of 'transfer'.

Response: We agree that transfer and transition are concepts that, while being related to each other, refer to different stages during the process of introducing the young person to adult care.

In order to avoid misinterpretations and clarify the difference between these concepts, we have added the definition of transfer in the background section (p.4).

-Lines 55-56 (p.4): "To achieve this..." this sentence appears to be unfinished.

Response: We have rephrased the sentence in order to clarify the meaning of it.

The sentence now states: "To increase the level of empowerment, a person-centered approach is important" (p.5).

-Lines 10-11 (p. 5): CHD may be a better acronym

Response: We have changed the acronym from ConHD to CHD in the entire manuscript.

-Lines 42/43 (p.5): "has a generic nature" maybe rephrase.

Response: In order to clarify the meaning of the sentence we have changed the wording.

Now the sentence reads as: "Although the project will encompass different chronic conditions, it will first study adolescents with CHD" (p.5).

-Methods/Design

Lines 47-57 (p.6) Does your sample size calculation consider potential drop-out. Would your sample size for the intervention group be large enough to take account of this? Thank you for raising this point.

Response: Although we had considered this, we did not mention it in the manuscript. Indeed, in order to compensate for possible dropouts, we have included an additional 10% of participants in our sample size calculation.

To make this clear for the reader, this section now reads as:

"In order to compensate for a potential 10% dropout rate, we will recruit 70 patients in each arm of the RCT. An additional 70 patients will be recruited in the centers of the control group. Among the four centers a total of 210 patients will be enrolled" (p.7).

-Lines 32/33 (p. 12) Using coloured ink in questionnaires has also been shown to increase responses rates. So if they look appealing this may encourage a response

Response: At the moment the logo of the project is the only section which is colored in the questionnaires. However, the reminders sent are printed using colored ink.

We appreciate the comment of the reviewer and will consider implementing this suggestion when we print new sets of questionnaires.

-Lines 32/33 the word “rate” could be removed

Response: We have corrected this (p.12).

-Lines 32-48 (p. 12) I wonder if the time to first follow-up of non-responders is a little long at three weeks? I think a one or two week gap might be better in terms of achieving a high response rate.

Also, as the project involves young people and mobile use is high in this group, a personalised text message or social media message may be an effective first reminder, that is not too intrusive.

Minimizing non-response in adolescent populations through the use of personalized text messages or social media is indeed an effective and easy way of communicating with the adolescents.

Response: We have included the suggestion from the reviewer on using text messages in the manuscript as a first reminder. We have also decreased the time between when the questionnaires are sent and the first reminder, to two weeks (p.12).

The sentence now states: “Two weeks after the documents are sent, non-responders will receive a first personalized reminder in the form of a text message” (p.12).

-Table 1. You use a mixture of child and adult questionnaires to measure outcomes. Are the questionnaires suitable for comparing young people and adults and are they all age appropriate at the time-points assessed? Do the participants fill out all questionnaires at each timepoint?

Indeed, the questionnaires are administered at each timepoint. This will allow us to calculate changes over time in all the measured variables.

Response: The questionnaires measuring knowledge and quality of life are the ones that have been used in adults. Nonetheless, the Knowledge scale for adults with congenitally malformed hearts (KnoCoMH) is based on the Leuven Knowledge Scale which has been validated in young persons with CHD as of the age of 12 years (Yang, 2012). This reference has been included in Table 1 (p.15). The linear analog scale validity has been evaluated in adolescents with CHD in the studies of Apers (2013) and Luyckx (2012). References to these articles have been provided in the article in Table 1 (p.16).

For the EQ-5D-3L there is a version that can be used in younger populations. However given the age of our participants, the adult version is the one that is recommended to use by the originators.

Overall, we can confirm that the questionnaires selected for this study can be used in young persons aged 16-18.5 years.

-Timing of transfer: Is this standardised across clinics? Maybe the methods sections should include information on the usual timing of transfer at each clinic and how variation could potentially affect the intervention. For example, if some patients didn't transfer until 20 years of age or they transferred at 16/17 years of age.

Response: As per intervention protocol, the age of transfer for patients of the intervention group will be transferred when they are 18 years.

It is standard practice in pediatric cardiology in Sweden to transfer patients around the age of 18 y.

Hence, the age of transfer in patients from the control and comparison group will be around the same age as the participants of the intervention group.

We have included the following section in the usual care description:

“It is common practice in Sweden that patients are transferred around the age of 18 years” (p.10).

In addition, we conduct usual care assessments throughout the project. This allows us to monitor if

practices in terms of age of transfer are different from the intervention group, or even if they are changing over time.

-P.18 -20 maybe some subheadings could be useful to break up the text in this section.

Response: We have added the subheadings in this section to separate the text (p.19-21).

-P.20 line 27 should be “transferred”

Response: We have corrected the typographical error (p.21).

P. 21 lines 5/6 “TC” and “DCO” are these acronyms necessary?

Response: The use of these two acronyms allows us to avoid repetition. If the editor deems it important to spell it out throughout the manuscript, we can certainly do so.

P.22 line 48 Should be: “will be submitted to peer reviewed journals for publication”

Response: We have rephrased the sentence following the suggestion of the reviewer (p.24).

Discussion

-P. 23 line 22 Is there an ‘and’ missing after the word ‘planning’?

Response: We have corrected the typographical error (p.24).

-P.23 line 37 – this should be ‘group’ rather than ‘groups’

Response: We have corrected the typographical error (p.25).

-P.23 line 55 ‘improved’ may be a better word than ‘increased’ here.

Response: We have rephrased the sentence following the suggestion of the reviewer (p.23).

-P.24 line 17 This should be ‘potential’ rather than ‘potentially’ I think

Response: We have corrected the typographical error (p.25).

-P.24 line 20 the end of this line should be rewritten perhaps ‘risk of biasing’ instead of ‘risk for bias’

Response: We have rephrased this sentence following the suggestion of the reviewer (p.25).

-P.25 line 44 I think the word ‘proves’ here is far too strong. Also the word ‘expected’ is too strong.

Possibly the sentence could be rephrased as:

“It is hoped the findings will provide strong evidence of the effectiveness of this intervention in young people with ConHD”

Response: We have rephrased this sentence following the suggestion of the reviewer (p.27).

-P.25 line 48 ‘can’ should be changed to ‘could potentially’.

Response: We have rephrased the sentence following the suggestion of the reviewer (p.27).

VERSION 2 – REVIEW

REVIEWER	Emily Heery Library and Research Service Oireachtas, Dublin Ireland
REVIEW RETURNED	21-Dec-2016

GENERAL COMMENTS	P. 12 (last paragraph) Following on my previous suggestion to
---

	provide the first reminder after two weeks instead of three, maybe the rest of the reminder schedule should be altered by one week (i.e. from 5 weeks to 4, 7 weeks to 6 and 9 weeks to 8). This should help to minimise non-response.
--	--

VERSION 2 – AUTHOR RESPONSE

As per request from the reviewer we have changed the time interval between each reminder, in an attempt to minimize non-response. The suggested changed can be found in page 12 of the manuscript.